# A counsellor-supported 'PTSD Coach' intervention versus enhanced Treatment-as-Usual in a resource-constrained setting: A randomised controlled trial

Erine Bröcker[1] , Miranda Olff[2] , Sharain Suliman[1] , Martin Kidd[3] , Lyrése Greyvenstein[1] and Soraya Seedat[1]

[1]Department of Psychiatry, Faculty of Medicine and Health Sciences, Stellenbosch University, Cape Town, South Africa; [2]Department of Psychiatry, Academic Medical Centre, University of Amsterdam, Amsterdam, Netherlands and [3]Centre for Statistical Consultation, Department of Psychiatry, Stellenbosch University

## Research Article

**Keywords:**
internet-based intervention; PTSD Coach mobile application; PTSD; adults; resource-constrained setting

**Corresponding author:**
Erine Bröcker;
Email: erineb@sun.ac.za

## Abstract

To widen treatment access for posttraumatic stress disorder (PTSD) in resource-constrained South Africa, we evaluated the feasibility and effectiveness of a counsellor-supported PTSD Coach mobile application (app) (PTSD Coach-CS) intervention on PTSD and associated sequelae in a community sample. Participants (female = 89%; black = 77%; aged 19–61) with PTSD were randomised to PTSD Coach-CS ($n = 32$) or enhanced Treatment-as-Usual ($n = 30$), and assessed with the Clinician-Administered PTSD Scale (CAPS-5), PTSD Checklist (PCL-5) and Depression, Anxiety and Stress Scale-21 items, at pre- to post-treatment and follow-up (1 and 3 months). We also collected data on user experiences of the PTSD Coach app with self-administered surveys. We conducted an intent-to-treat analysis and linear mixed models. A significant (group × time) effect for the CAPS-5 ($F_{3.136} = 3.33$, $p = 0.02$) indicated a greater reduction in PTSD symptom severity over time for the intervention group with a significant between-group effect size detected at 3-month follow-up. Significant between-group effect sizes were detected in self-reported stress symptom reduction in the intervention group at post-treatment and 3-month follow-up. Participants perceived the app as helpful and were satisfied with the app. Findings suggest PTSD Coach-CS as a suitable low-cost intervention and potential treatment alternative for adults with PTSD in a resource-constrained country. Replication in larger samples is needed to fully support effectiveness. Pan African Trial Registry: PACTR202108755066871.

## Impact statement

Many individuals in low-middle-income countries (LMICs), such as South Africa, often experience high trauma exposure rates and associated posttraumatic stress disorder (PTSD). PTSD can have harmful effects on the individual with their daily functioning often negatively impacted. Unfortunately, many individuals with PTSD in these settings do not access the needed support and treatment due to healthcare resource constraints. These resource constraints include overburdened public health care services and difficulty accessing these services when available. More feasible intervention alternatives are needed to widen access to support and treatment in the general population. Mobile-based interventions, such as the freely available PTSD Coach mobile application (app), is one such alternative. We evaluated the effectiveness of a four-session counsellor-supported PTSD Coach app (PTSD Coach-CS) intervention in reducing PTSD, depression, anxiety, and stress symptoms in a South African adult community sample. Our findings support that a low-cost, more accessible PTSD Coach-CS intervention is a feasible intervention alternative for adults with PTSD in a resource-constrained setting such as South Africa. It also appears that it can reduce PTSD and stress symptoms; however, more research is needed to fully support this. Importantly, our findings indicate that the original version of the app, developed for veterans in the United States, can be used in a culturally different setting and that intervention delivery can be supplemented with counsellors to increase intervention engagement. This was the first time the PTSD Coach app was evaluated in an LMIC setting and was different in supplementing delivery with less specialised mental health services (registered counsellors and not psychologists). These findings will hopefully lead to improved access to care for those with PTSD in a South African resource-constrained setting.





## Introduction

Low- and middle-income countries (LMICs) are characterised by high prevalence rates of trauma exposure and associated PTSD (Kessler et al., 2017; Koenen et al., 2017; Seedat and Suliman, 2018; Ng et al., 2020; Hiscox et al., 2021; Kayiteshonga et al., 2022; White et al., 2023). Additionally, the extent of psychiatric comorbidity in PTSD, especially depressive and anxiety disorders, is well-documented (Rytwinski et al., 2013; Morina, 2018; Kayiteshonga et al., 2022). This disease burden coupled with widespread healthcare constraints in many LMICs results in many individuals with PTSD not accessing adequate care (Docrat et al., 2019; Knettel et al., 2019).

In South Africa, an LMIC, trauma exposure, PTSD, and its associated sequelae are indeed major public health concerns, exacerbated by significant healthcare resource constraints (Atwoli et al., 2013; Benjet et al., 2016; Seedat and Suliman, 2018; Docrat et al., 2019; Okafor et al., 2021). These resource constraints include inadequate and overburdened services, especially in the public health care sector, and the lack of accessibility to these services (Docrat et al., 2019; Independent Communications Authority of South Africa [ICASA], 2020). Barriers to the accessibility of available resources often include financial and time constraints (e.g., travel costs and time), while the stigma associated with PTSD and psychiatric care itself further hinders treatment accessibility (Booysen et al., 2021; Monnapula-Mazabane and Petersen, 2021). Thus, more feasible, affordable and suitable intervention alternatives are thus needed to widen access to public mental health services in the general South African population (Seedat and Suliman, 2018).

With increased internet access, freely available mental health-focused mobile-based interventions, are one alternative to broaden access to psychiatric care and overcome barriers to treatment seeking (Olff, 2015; Sander et al., 2020). Mental health-focused mobile-based interventions hold promise as they provide greater capacity for support, are available at hand and immediately, can be anonymous and private, and appear cost-effective (Olff, 2015; Ruzek and Yeager, 2017; van der Meer et al., 2020). Mobile-based interventions pose a plausible option since most of South Africa has cellular coverage, with about 65 million active smartphone subscriptions (Olff, 2015; Ruzek and Yeager, 2017; ICASA, 2020; Sander et al., 2020). Specifically, the freely available PTSD Coach mobile application (app) may address some of the above treatment barriers and increase access for adults with PTSD in the South African public healthcare setting (Hoffman et al., 2011; Sander et al., 2020).

The PTSD Coach app was released by the United States' Department of Defence and Veteran Affairs in 2011, with an updated version launched in 2018 (Hoffman et al., 2011; Kuhn et al., 2018). This self-managed app intervention aims to assist trauma-exposed adults with PTSD psychoeducation, encourage them to seek treatment if so indicated, and monitor and manage their PTSD symptoms (Hoffman et al., 2011; Kuhn et al., 2018). The following key characteristics make the PTSD Coach app an appealing treatment option in resource-constrained settings: freely available with no cost associated post-download (e.g., in-app purchases); not requiring an internet connection post-download; and utilising limited phone memory (130 megabytes) (Kuhn et al., 2018). Additional benefits include enhancements for visual and hearing impairments and existing evidence-based research that continues to expand (Kuhn et al., 2018). App features are divided into four core functions: (1) 'Learn', (2) 'Track progress', (3) 'Manage symptoms' and (4) 'Get support.' The first function focuses on psychoeducation about PTSD symptoms, while the second includes methods for individual symptom tracking and feedback on progress made. The 'Manage symptoms' function encompasses eight core PTSD symptoms with 21 accompanying symptom management tools. These tools encompass relaxation, mindfulness and stress management exercises to assist with the management of distress associated with PTSD. Since the app is not intended to replace professional mental health care, the last function focuses on learning about and identifying professional treatment services (i.e., mental health providers). It is also possible to customise this 'Get support' function by adding emergency contacts and saving preferred resources. The PTSD Coach app has received positive quality ratings (i.e., engagement, functionality, aesthetics and information) and adheres to strict data protection and privacy standards (Hoffman et al., 2011; Sander et al., 2020). A review of 69 available PTSD-focused apps found that the PTSD Coach app was the highest rated, with several feasibility, acceptability and effectiveness data (Sander et al., 2020).

Trials of the PTSD Coach app, to date, have been conducted in high-income countries with high smartphone ownership and more mental health resources, and have indicated positive outcomes in the intervention group (Miner et al., 2016; Possemato et al., 2016; Kuhn et al., 2017, 2018; Pacella-Labarbara et al., 2020; van der Meer et al., 2020; Hensler et al., 2022). However, the results of a meta-analysis of randomised controlled trials (RCTs) evaluating the effectiveness of a PTSD Coach app intervention on self-reported posttraumatic stress symptom (PTSS) reduction indicated that the pooled effect size was not significant at post-treatment (Bröcker et al., 2023). Results from pre- to post-treatment studies also indicated a non-significant decrease in self-reported PTSS symptoms (Cernvall et al., 2018; Tiet et al., 2019). Generally, research to date suggests more evidence of the effectiveness of the PTSD Coach app intervention is needed, specifically in LMICs.

Furthermore, while the app was designed for self-management use, research on PTSD Coach and other mental health-focused apps suggests that adding a supportive component to intervention delivery may enhance intervention engagement and effectiveness (Possemato et al., 2016; Pacella-Labarbara et al., 2020; Rodriguez et al., 2021). Specifically, prior findings and recommendations from PTSD Coach app research that supplemented intervention delivery with virtual support informed the addition of an in-person supportive component (Bröcker et al., 2023). Supplementing intervention delivery with less specialised mental health services (i.e., registered counsellors) under supervision can address the need for upscaling and task shifting of public mental health care services in South Africa to widen access to care (Keynejad 2018; Spedding et al., 2015; Rossouw et al., 2018). The category of a registered counsellor was established in South Africa to increase accessibility to mental health care specifically at the community level (i.e., community clinics) (Health Professions Council of South Africa [HPCSA], 2022). Compared to more specialist providers (e.g., psychologists or psychiatrists), HPCSA-registered counsellors receive 4 years of training, inclusive of a practicum in a community setting. Their scope of practice allows them to work alongside specialist providers, and their skillset includes an understanding of psychopathology, the ability to effectively screen and refer when necessary, risk monitoring (i.e., suicidality) and the provision of supportive short-term psychological and preventative interventions (HPCSA, 2022). In the context of supportive mental health-focused mobile-based interventions, registered counsellors can offer support in working through a structured manual-guided intervention with a patient while under the supervision of specialist providers.

Considering the above, we conducted an RCT evaluating the effectiveness of a brief four-session counsellor-supported PTSD

Coach app (PTSD Coach-CS) intervention compared with enhanced Treatment-as-Usual (e-TAU) in adults with PTSD in a South African community sample. We hypothesised that PTSD Coach-CS is superior to e-TAU in reducing clinician-monitored PTSD symptoms (primary outcome) as well as self-reported PTSD, depression, anxiety stress and stress symptoms (secondary outcomes). We also report on treatment engagement and perceived helpfulness and satisfaction of the PTSD Coach app.

## Methods

The trial was prospectively registered in the Pan African Clinical Trials Registry (PACTR202108755066871) and was approved by the Health Research Ethics Committee, Stellenbosch University (SU) (N18/10/132; S18/05/058). All trial data were anonymised to protect participants' privacy and to ensure confidentiality. We used the CONSORT checklist when writing our report (Schulz et al., 2010).

### Design

In this single-blind, parallel-arm RCT participants were allocated to four sessions of PTSD Coach-CS or e-TAU. An independent evaluator (IE), a qualified clinical psychologist who was blinded to intervention allocation, monitored all participants for treatment response from pre- (T1), to post-treatment at 4 weeks (T2), 1-month follow-up (T3) and 3-month follow-up (T4).

### Participants

Trauma-exposed adults (18–65 years) with a current PTSD diagnosis as confirmed by the Mini-International Neuropsychiatric Interview for DSM-5 (MINI 7.0.2) with moderate and above symptom severity (as per the CAPS-5 total score of ≥23) were eligible (Sheehan et al., 1998; Weathers et al., 2018). Further eligibility criteria included: (i) smartphone ownership; (ii) conversant in English since we used the original PTSD Coach app; (iii) ability to attend study visits at the Faculty of Medicine and Health Sciences (FHMS), SU; and (iv) provide written informed consent. Subject to the MINI 7.0.2 results, participants were excluded in the presence of (i) current (past 6 months) substance use disorder; (ii) high suicidal risk; and/or (iii) cognitive impairment. These exclusions were related to the possible compromised ability to engage with the intervention. Participants who were receiving psychotherapy or who had a recent change in psychotropic medications (≤2 months) were also excluded.

Our target sample size was based on Miner et al.'s (2016) study which compared a PTSD Coach app intervention with waitlisted control in a community sample. They indicated that $N = 120$ would be sufficient to detect statistically significant between-group effect sizes at post-treatment. Thus, our target sample was powered ($\alpha = .05$; power $= .80$) to detect a medium effect size between intervention ($n = 60$) and control ($n = 60$) at post-treatment.

### Procedure

The study was promoted with flyers and information sheets distributed via social media and in the community. This included sharing study information with interested clinicians through WhatsApp and placing flyers at local police stations and community clinics. Participants were screened telephonically with a custom-designed eligibility questionnaire and the Global Psychotrauma

Screen (GPS) (Olff et al., 2021) (see Supplementary material S1). Potential participants with probable PTSD, as determined by the GPS (PTSD subdomain score ≥3/5), were invited for a pre-treatment assessment. The pre-treatment assessment at baseline (T1) entailed obtaining written informed consent, completing a demographic questionnaire, self-report measures and diagnostic assessments (MINI 7.0.2 and CAPS-5).

Enrolled participants were handed over to the counsellor for intervention allocation and study visit arrangements (i.e., intervention visits and follow-ups), and excluded participants were appropriately referred. The IE provided a referral letter for each participant (see *Interventions: e-TAU*). The follow-up assessments entailed the CAPS-5 to monitor the primary outcome and self-reports related to the secondary outcomes. Participants were not financially compensated but received transport cost reimbursement (ZAR200/USD13.30)[1] and refreshments at each study visit. At study conclusion (T4), participants received an airtime voucher (ZAR30/USD1,60) to thank them for their time.

### Randomisation

Participants were randomised (1:1 ratio) to PTSD Coach-CS or e-TAU using a computerised allocation sequence generated by an independent researcher. Allocations sealed in envelopes labelled with corresponding participant numbers (e.g., PTSD001) were stored securely by the counsellor, who opened them upon enrolment of participants.

### Baseline data

At baseline (T1), sociodemographic (age, sex, educational level, marital status, employment and annual income bracket) and clinical characteristics (i.e., childhood trauma, lifetime and index trauma exposure, perceived social support, resilience, substance use, past psychiatric care and current psychiatric comorbidity) were collected to identify and control for potential covariates known to influence PTSD severity and/or treatment response (Kessler et al., 2017; Catabay et al., 2019; Blais et al., 2021; Nöthling et al., 2022) (see Supplementary material S1 for further details).

The 17-item Life Events Checklist for DSM-5 (LEC-5) captured lifetime trauma exposure and identified an index trauma (T1) (Weathers et al., 2013). The LEC-5 was revisited at each visit (T2–T4) to capture additional trauma exposure and participants were reminded to endorse symptoms on the CAPS-5 and PCL-5 based on their initially identified index trauma. A summary of the outcome measures and applicable Cronbach's alpha (α) values for the present sample follows.

### Outcome measures

The CAPS-5, PCL-5 and DASS-21 were administered at all visits (T1–T4). The perceived helpfulness of the PTSD Coach app survey and the self-efficacy managing PTSD symptoms were administered upon intervention completion.

#### Primary outcome measure
Clinician-administered PTSD Scale for DSM-5 (CAPS-5), past month version. The CAPS-5 total score assessed change in clinician-monitored PTSD symptom severity. This 30-item structured diagnostic interview assessed the frequency and severity of

---

[1]Exchange rates in June 2023 used for all currency conversion calculations.

PTSD symptoms as per DSM-5 over the past month in response to the index trauma (Weathers et al., 2018). The CAPS-5 demonstrated good internal consistency (Cronbach's α = 0.87) in our sample.

### Secondary outcome measures
PTSD checklist for DSM-5 – PCL-5.  The PCL-5 total score assessed change in self-reported PTSD symptom severity. This 20-item self-report measure evaluated the degree (0 = '*Not at all*' to 4 = '*Extremely*') to which participants were bothered by PTSD symptoms as per DSM-5 in the past month (Wortmann et al., 2016). In our sample, a clinical cut-off score of ≥33 was used as indicative of probable PTSD (Verhey et al., 2018). The PCL-5 demonstrated high internal consistency (Cronbach's α = 0.96) in the present sample.

Depression, anxiety and stress scale 21 – DASS-21.  The DASS-21 sub-scale scores assessed change in self-reported depression, anxiety and stress symptom severity (Lovibond and Lovibond, 1995; Henry and Crawford, 2005). Respective clinical cut-off scores are: DASS-Depression (≤13); DASS-Anxiety (≤9) and DASS-Stress (≤18) (Lovibond and Lovibond, 1995). The scales demonstrated good to high internal consistency in the present sample: DASS-Depression (Cronbach's α = 0.93); DASS-Anxiety (Cronbach's α = 0.87) and DASS-Stress (Cronbach's α = 0.91).

Perceived helpfulness of the PTSD Coach app survey.  PTSD Coach-CS participants completed this survey that evaluated their perceived helpfulness of the app through 14 items rated on a 4-point scale (0 = '*Not at all helpful*' to 4 = '*Extremely helpful*') (Kuhn et al., 2014). The survey concludes with an item evaluating the user's overall satisfaction with the app with answers ranging from 0 = '*Not at all satisfied*' to 4 = '*Extremely satisfied*'. For the present sample, the 14 items measuring the perceived helpfulness of the app demonstrated high internal consistency (Cronbach's α = 0.93).

Self-efficacy managing PTSD symptoms.  This 10-item scale evaluated the participants' confidence in managing their PTSD-related symptoms (Kuhn et al., 2017). Responses were measured on a scale from 0 to 100, with 0 = '*Cannot do at all*', 50 = '*Moderately can do*' and 100 = '*Highly certain can do*'. The scale demonstrated good internal consistency (Cronbach's α = 0.84).

### Interventions

All participants attended intervention sessions in person at the FMHS (Stellenbosch University medical school), SU.

#### PTSD Coach-CS
Participants attended four weekly counsellor-supported sessions of 30–40 min each. The registered counsellor was trained in a standard support method (assisting with language and technology difficulties and not providing therapeutic support) and to adhere to the study-designed treatment protocol (see Table 1). Restricting counsellor involvement (i.e., not providing therapeutic support) was informed by sensitivity towards the scope of practice delineated for registered counsellors (HPCSA, 2022). The PTSD Coach app was downloaded onto participants' smartphones, with mobile data provided by the study. All sessions followed the same structure: (i) setting an agenda for the session; (ii) gathering feedback on the last session; (iii) reviewing the week (i.e., enquiring about the frequency of app use during the week) and homework (i.e., using app tools); (iv) accessing and using selected symptoms and tools,

with the counsellor available to assist with language and technical difficulties and (v) agreeing on homework for the following week (i.e., identifying tools to explore more). These tools were selected after carefully considering their suitability and appropriateness to our setting. For example, the tool and accompanying activities under 'Leisure Activities' were excluded since they can pose complications for some in our setting due to safety concerns (or were restricted due to pandemic-related restrictions at the time). Participants were encouraged to use the app outside of the intervention sessions and to review the material covered in the session.

#### e-TAU
Participants received a detailed referral letter, including a symptom profile and a request for psychological assistance at the primary health care level. The counsellor aimed to contact the local clinic, inform them of the referral, and arrange participant appointments. Participants received a list of non-governmental organisations and mental health helplines as additional support resources. Participants were also encouraged to contact the counsellor should they need support. Hereafter, participants were followed up by the counsellor regarding treatment accessed after 4 weeks (T2) and 8 weeks (T3) before the monitoring visits. These follow-ups tracked symptom levels, risk profile (i.e., suicidality), if treatment was sought and/or received, and what it entailed.

#### Treatment fidelity
Treatment fidelity was ensured by (i) training of the counsellor in the PTSD Coach-CS protocol and e-TAU procedures, and (ii) a review of case notes after the trial concluded. The PTSD Coach-CS and e-TAU case notes required completion at each visit, documenting protocol adherence and other feedback from participants and the counsellor. A research team member (S. Suliman) was available to the counsellor for support (i.e., study procedure uncertainty or risk management) to ensure that the IE remained blinded to intervention allocation.

### Data analysis

Data were analysed using SPSS (descriptive statistics) and Statistica (R package 'ImerTest' version 3.1-0 for mixed models and 'geepack' version 1.2-1 for the generalised estimating equation [GEE] analysis) (Dell, 2014; IMB Corp, 2017).

We conducted intent-to-treat analysis including all participants in the statistical analysis focusing on both statistical (one-tailed alpha of 0.05) and clinical significance (clinical cut-off) of group differences over time (T1–T4). Baseline characteristics were compared between the arms using *t* tests (continuous variables) or chi-square (categorical variables) to assess whether the intervention arms were balanced through randomisation and to identify covariates. Since there were no significant baseline differences, no covariates were included in the analysis.

Linear mixed models (LMMs), with participants as random effects and group, time and (group × time) as fixed effects were used to assess the effectiveness of the intervention on primary and secondary outcomes and compared between and within-group differences. The Fisher's least significant difference test was used as a post hoc test to identify statistically significant mean differences between groups at each time point. Based on the smaller sample size, the Hedge's *g* statistic was used to measure effect sizes. GEEs were used to assess between-group differences over time in the percentage of participants meeting the clinical cut-offs of the CAPS-5, PCL-5 and DASS-21 subscales at each time point. Both

**Table 1.** Intervention content

| Session 1 | | |
|---|---|---|
| Dashboard | Tools | Activity |
| Learn | About PTSD | What is PTSD? |
| | Getting professional help | How does PTSD develop? Is counselling confidential? |
| | PTSD and the family | Fighting fair |
| Manage symptoms | Reminded of the trauma | Body scan with Julia Deep breathing |
| | Avoiding triggers | RID: Coping with triggers Observe thoughts |
| **Session 2** | | |
| Dashboard | Tools | Activity |
| Manage symptoms | Disconnected from people | Connect with others Relationship tools |
| | Disconnected from reality | Grounding Muscle relaxation |
| **Session 3** | | |
| Dashboard | Tools | Activity |
| Manage symptoms | Sad/hopeless | Seeing my strength Change your perspective |
| | Worried/anxious | Mindful breathing Thought shifting |
| **Session 4** | | |
| Dashboard | Tools | Activity |
| Manage symptoms | Angry | Mindfulness Positive imagery |
| | Unable to sleep | Good sleep habits Ambient sounds |

LMM and GEE accommodate missing data (loss to follow-up assessments).

## Results

### Participant flow

Enrolment started in November 2021 and was terminated in January 2023, with follow-ups completed in May 2023. Sixty-two participants were enrolled and randomised to PTSD Coach-CS ($n = 32$) or e-TAU ($n = 30$) (see Figure 1). Study retention was 82% at post-treatment, 73% at 1-month follow-up and 66% at 3-month follow-up. Most participants lost to follow-up were in the e-TAU arm.

### Baseline characteristics

Tables 2 and 3 provide the baseline demographic and clinical characteristics. Participants ($N = 62$) ranged between 19 and 61 years ($M = 34.40$, $SD = 11.35$) were predominantly female (89%), self-identified as black (77%) and reported isiXhosa (69%) as their first language. Over half (60%) were unemployed, 65% reported an annual income below ZAR10 000/USD550 and 52% did not complete secondary education.

Of those who self-reported previous psychiatric diagnoses ($n = 27$; 44%), a few ($n = 4$) were previously diagnosed with PTSD and received medication ($n = 2$) or counselling ($n = 2$) at the time of diagnosis. Twenty-four percent of the total sample were on psychotropic medication at trial entry and none were receiving psychological support (i.e., counselling).

Current psychiatric co-morbidity was high (87%) with the most common comorbid disorder being major depressive disorder (MDD) (52%). The most widely endorsed index trauma exposures were interpersonal violence (48%), loss and illness (40%) and serious motor vehicle accidents (12%).

There were no significant baseline between-group differences for all demographic and clinical variables. Significant differences between those who were lost to follow-up and those who were not at post-treatment included: employment ($p = 0.03$), annual income ($p = 0.03$) and psychiatric comorbidity of MDD, past ($p = 0.04$). Those who were lost to follow-up were more likely to be employed ($n = 8$; 72%), have an annual income >ZAR10 000/USD530 ($n = 4$; 36%), and meet diagnostic criteria for MDD past ($n = 7$; 63%).

### Engagement

#### PTSD Coach-CS (n = 32)

The counsellor successfully downloaded the app for 23 (72%) participants during the first intervention visit. Reasons for unsuccessful downloads for the remainder included technical difficulties (data was given to the participant so they could try to download the app at home) and phone memory difficulties (the counsellor provided memory cards to participants) (see Table 4). App store accessibility remained problematic for one participant, and a study phone was used during the intervention sessions. The remaining participants attended all intervention sessions, with one missing the last session due to work responsibilities. During the 4-week intervention period, self-reported app use outside the sessions varied from daily to five times per week.

#### e-TAU (n = 30)

The counsellor successfully scheduled appointments (i.e., mental health services) for six (20%) participants at their respective local clinics and established contact and awaited appointment confirmation for 11 (37%) participants at first contact. She could not establish contact with the respective clinics for 11 (37%) participants, with two (7%) declining linkages to care at their local clinic. After 4 weeks, five (17%) received support, an additional three (10%) received appointment dates, while 14 (47%) participants still awaited linkages to care at their local clinic. The remainder ($n = 7$; 23%) were lost to follow-up. After 8 weeks, a total of eight (36%) received support, another one (5%) received an appointment date and six (27%) were still awaiting linkages to care at their local clinic. The remainder ($n = 4$; 18%) were lost to follow-up.

### Intervention outcomes

#### Primary outcome

Clinician monitored PTSD. There was a significant interaction effect (group × time) for the primary outcome, CAPS-5 total score ($F_{3.136} = 3.33$, $p = 0.02$), indicating a greater reduction in PTSD symptom severity over time for the intervention versus control group (see Figure 2). However, at post-treatment the between-group effect size was not significant (Hedges $g = 0.33$, $p = 0.22$) with a similar trend detected at 1-month follow-up (Hedges $g = 0.37$, $p = 0.10$). A large effect size with between-group

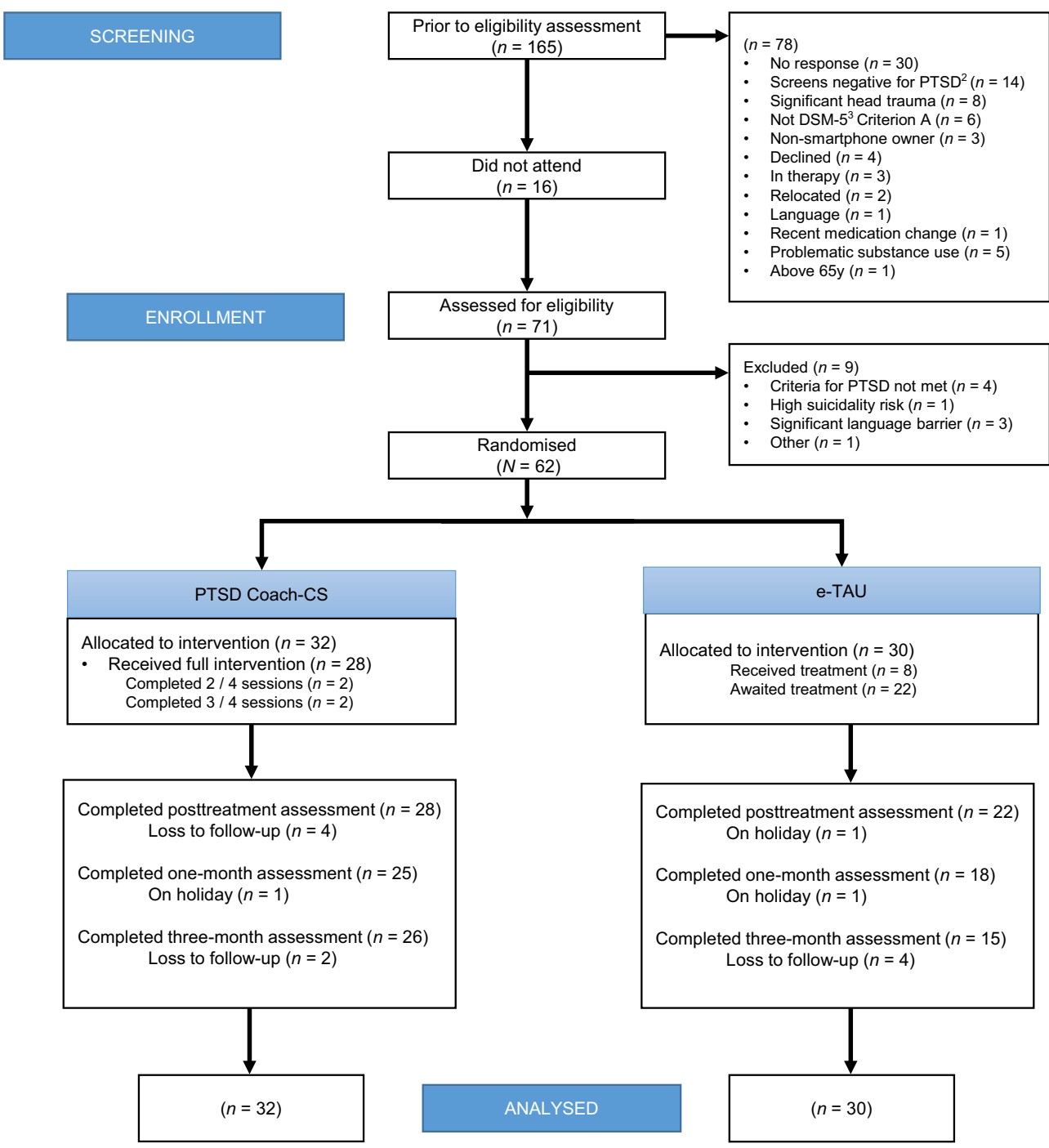

**Figure 1.** Consort flow diagram.

significance was detected at the 3-month follow-up (Hedges *g* = 0.86, *p* = <0.01).

When considering the CAPS-5 cut-off for PTSD (<23), there was no significant group × time interaction effect (Wald = 1.51, *p* = 0.68) (see Table 5). The proportion of participants falling below the clinical cut-off was higher but not significantly so in the intervention group compared to the control group at follow-up (37% vs. 20%, *p* = 0.24), and 1-month (56% vs. 30%, *p* = 0.17). At the 3-month follow-up the difference was significant (69% vs. 27%, *p* = 0.05).

**Table 2.** Baseline socio-demographic characteristics

| Variable | Total sample (n = 62) n (%)/mean (SD) | PTSD Coach-CS (n = 32) n (%)/mean (SD) | e-TAU (n = 30) n (%)/mean (SD) | p |
|---|---|---|---|---|
| Age (years) | | | | 0.31 |
| | 37.40 (11.35) | 36.63 (11.98) | 38.23 (10.78) | |
| Sex | | | | 0.26 |
| Female | 55 (89%) | 27 (84%) | 28 (93%) | |
| Male | 7 (11%) | 5 (16%) | 2 (7%) | |
| Ethnicity | | | | 0.71 |
| Black | 48 (77%) | 26 (81%) | 22 (73%) | |
| Mixed-race[1] | 11 (18%) | 5 (16%) | 6 (20%) | |
| White | 3 (5%) | 1 (3%) | 2 (7%) | |
| First language | | | | 0.21 |
| Xhosa[2] | 43 (69%) | 23 (72%) | 20 (67%) | |
| Afrikaans[2] | 10 (16%) | 3 (9%) | 7 (23%) | |
| English | 7 (11%) | 5 (16%) | 2 (7%) | |
| Sesotho[2] | 1 (2%) | – | 1 (3%) | |
| Shona[3] | 1 (2%) | 1 (3%) | – | |
| Highest level of education | | | | 0.84 |
| Grade 9 | 2 (3%) | 2 (6%) | – | |
| Grade 10 | 4 (6%) | 2 (6%) | 2 (7%) | |
| Grade 11 | 12 (21%) | 6 (19%) | 7 (23%) | |
| Grade 12 | 30 (48%) | 15 (47%) | 15 (50%) | |
| Tertiary | 13 (21%) | 7 (22%) | 6 (20%) | |
| Marital status | | | | 0.88 |
| Single | 32 (52%) | 18 (56%) | 14 (47%) | |
| Married/cohabiting | 21 (34%) | 11 (34%) | 10 (33%) | |
| Divorced/separated | 5 (8%) | 2 (6%) | 3 (10%) | |
| Widowed | 4 (6%) | 1 (3%) | 3 (10%) | |
| Employment | | | | 0.42 |
| Unemployed | 37 (60%) | 20 (63%) | 17 (57%) | |
| Employed | 24 (38%) | 11 (34%) | 13 (43%) | |
| Pensioner | 1 (2%) | 1 (3%) | – | |
| Annual income | | | | 0.72 |
| <ZAR10 000[4] | 40 (65%) | 20 (63%) | 20 (67%) | |
| ZAR10–20000[5] | 5 (8%) | 2 (6%) | 3 (10%) | |
| ZAR20–40000[6] | 4 (6%) | 3 (9%) | 1 (3%) | |
| ZAR40–60000[7] | 5 (8%) | 3 (9%) | 2 (7%) | |
| ZAR60–100000[8] | 3 (5%) | 2 (6%) | 1 (3%) | |
| >ZAR100 0009 | 5 (8%) | 2 (6%) | 3 (10%) | |

*Note*: SD, standard deviation; [1]Self-identified as belonging to the mixed ancestry ethnic group; [2]One of the main languages in South Africa; [3]One of the main languages in Zimbabwe; [4]USD < 540; [5]530–1,060; [6]1060–2,120; [7]2120–3180; [8]3180–5300; [9] > 5300.

### Secondary outcomes

**Self-reported PTSD.** There was a larger reduction in self-reported PTSD symptom severity (PCL-5) in the intervention group compared to the control group at all time points; however, the interaction effect (group × time) over time was not significant ($F_{3.135} = 1.21$, $p = 0.31$) (see Supplementary Figure S1). The proportion of participants in the intervention group compared to the control group meeting the cut-off score (<33) was higher but not

**Table 3.** Baseline trauma exposure and clinical characteristics

| Variable | Total sample (*n* = 62) n (%)/mean (SD) | PTSD Coach-CS (*n* = 32) n (%)/mean (SD) | e-TAU (*n* = 30) n (%)/mean (SD) | *p* |
|---|---|---|---|---|
| Previous psychiatric diagnosis (lifetime)[1] | | | | 0.20 |
| Yes | 27 (44%) | 11 (34%) | 16 (53%) | |
| No | 35 (56%) | 21 (66%) | 14 (47%) | |
| Previous psychiatric treatment (lifetime)1 | | | | 0.14 |
| Yes | 33 (53%) | 14 (44%) | 19 (63%) | |
| No | 29 (47%) | 18 (56%) | 11 (37%) | |
| Previous psychiatric medication (lifetime)1 | | | | 0.07 |
| Yes | 23 (37%) | 8 (25%) | 15 (50%) | |
| No | 39 (63%) | 24 (75%) | 15 (50%) | |
| Current psychiatric medication1 | | | | 0.38 |
| Yes | 15 (24%) | 6 (19%) | 9 (30%) | |
| No | 47 (76%) | 26 (81%) | 21 (70%) | |
| Current or past psychiatric comorbidity[2] | 55 (89%) | 29 (91%) | 26 (87%) | 0.46 |
| MDD, Current | 31 (50%) | 18 (56%) | 13 (43%) | |
| MDD, Past episode | 22 (35%) | 11 (34%) | 11 (37%) | 0.18 |
| Bipolar Mood Disorder, Type II | 1(2%) | – | 1 (3%) | |
| Anxiety disorders | 16 (26%) | 7 (22%) | 8 (27%) | 0.53 |
| Agoraphobia | 5 (8%) | 2 (6%) | 3 (10%) | |
| Generalised Anxiety Disorder | 3 (5%) | 2 (6%) | 1 (3%) | |
| Social anxiety disorder | 4 (6%) | 1 (3%) | 3 (10%) | 0.77 |
| Panic disorder, lifetime | 2 (3%) | 2 (6%) | – | |
| Panic disorder, current | 1 (2%) | – | 1 (3%) | |
| CTQ-SF | 55.48 (19.43) | 57.56 (21.56) | 53.27 (17.19) | 0.62 |
| RES | 24.85 (6.20) | 25.28 (7.02) | 24.40 (5.27) | 0.58 |
| MSPSS | 57.73 (15.88) | 58.56 (16.50) | 56.83 (15.43) | 0.67 |
| AUDIT* | 5.00 (7.43) | 4.34 (6.99) | 5.70 (7.92) | 0.47 |
| LEC-5 | | | | |
| Mean no, of traumas (experienced) | 4.73 (2.26) | 4.97 (2.40) | 4.47 (2.10) | 0.38 |
| Mean no. of traumas (witnessed) | 3.06 (2.03) | 3.03 (2.16) | 3.10 (1.92) | 0.89 |
| Index trauma | | | | |
| Interpersonal violence[3] | 30 (48%) | 17 (53%) | 13 (43%) | 0.62 |
| Loss or illness[4] | 25 (40%) | 11 (34%) | 14 (47%) | |
| Accident (MVA) | 7 (11%) | 4 (12%) | 3 (10%) | |

*Note*: [1]Self-reported; [2]Assessed with the MINI 7.0.2; AUDIT, alcohol use disorders identification test; CTQ-SF, childhood trauma questionnaire—short form; LEC-5, life events checklist for DSM-5; MSPSS, multidimensional scale of perceived social support; MVA, motor vehicle accident; RES, resilience evaluation scale; SD, standard deviation. *Participants denied substance use other than alcohol thus Drug Use Disorders Identification Test results not reported. [3]Sexual assault, physical assault, assault with a weapon. [4]Traumatic loss of loved one, life-threatening injury or illness.

significant at post-treatment (50% vs. 37%. *p* = 0.58), 1-month (62% vs. 37%. *p* = 0.18) and 3-month follow-up (72% vs. 33%, *p* = 0.13).

*Self-reported depression, anxiety and stress.* There was a significant interaction effect (group × time) in DASS-Stress scores between groups over time ($F_{3.136}$ = 3.82, *p* = 0.01), indicating a greater stress reduction for the intervention group (see Supplementary Figure S4). The between-group effect size was significant at post-treatment (Hedges *g* = 0.50, *p* = 0.02) and 3-month follow-up (Hedges *g* = 0.71, *p* = 0.03) (see Table 5). At 1-month follow-up, a trend towards significance was observed (Hedges g = 0.01, *p* = 0.08). A significantly larger proportion of participants in the intervention group compared to the control group met the cut-off score at post-treatment (69% vs. 37%. *p* = 0.02), larger but not significantly so at 1-month (59%

**Table 4.** PTSD Coach-CS treatment engagement

| Elements | Answer | Session 1 (n = 32) n (%) | Session 2 (n = 30) n (%) | Session 3 (n = 30) n (%) | Session 4 (n = 28) n (%) |
|---|---|---|---|---|---|
| App download | Yes | 23 (78%) | 29 (91%) | 29 (91%) | 29 (91%) |
| | No | 9 (28%) | 1 (9%) | 1 (9%) | 1 (9%) |
| | Unable | 5 (16%) | – | – | – |
| | Phone memory | 2 (6%) | – | – | – |
| | Low battery | 1 (3%) | – | – | – |
| | App store access | 1 (3%) | 1 (3%) | 1 (3%) | 1 (3%) |
| Attended | Yes | 32 (100%) | 30 (94%) | 30 (94%) | 27 (85%) |
| | No | 0 (0%) | 2 (6%) | 0 (0%) | 3 (9%) |
| | Loss to follow-up | – | 2 (6%) | – | 2 (6%) |
| | Could not attend | – | – | – | 1 (3%) |
| App use frequency over the past week (self-reported) | Used | NA | 21 (70%) | 21 (76%) | 24 (86%) |
| | Daily | | | | |
| | One to two times per week | | 4 (13%) | 9 (30%) | 9 (32%) |
| | Three to four times per week | | 10 (33%) | 10 (33%) | 4 (14%) |
| | Five times per week | | 7 (23%) | 8 (27%) | 7 (25%) |
| | | | – | 2 (7%) | 4 (14%) |
| | Did not use | NA | 10 (33) | 1 (3%) | 4 (14%) |
| | Unspecified | | 1 (3%) | – | 2 (7%) |
| | No access | | 9 (30%) | 1 (3%) | 1 (3%) |
| | Phone stolen | | – | – | 1 (3%) |

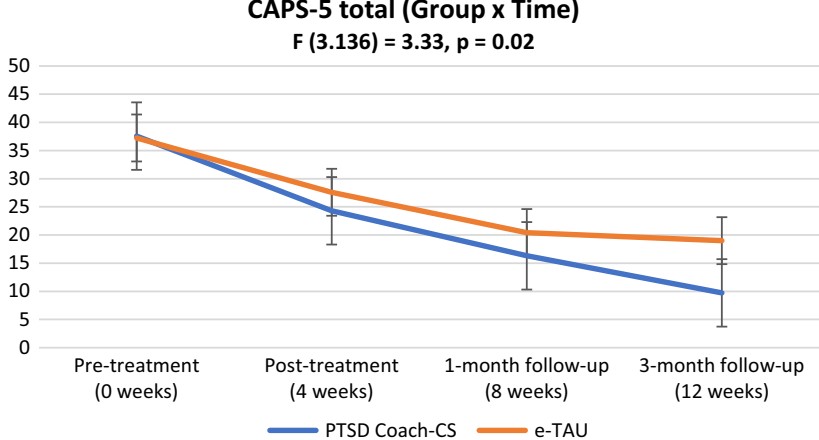

**Figure 2.** CAPS-5 total score: Interaction over time.

vs. 43%. $p = 0.47$), and at 3-month follow-up (69% vs. 33%. $p = 0.26$). However, when considering the DASS-Stress cut-off as an outcome, there was no significant group × time interaction effect (Wald = 3.28, $p = 0.35$). While there was a larger reduction in both DASS-Depression and DASS-Anxiety scores in the intervention group compared to the control group at all time points, no significant interaction effects (group × time) or in clinical cut-off scores were observed for either subscale (see Table 5, Supplementary Figures S2 and S3).

*Helpfulness and satisfaction.* Results indicated that participants perceived the PTSD Coach app as moderately to very helpful and

**Table 5.** Between-group comparison of the outcome variables at each time-point

| Variable | PTSD Coach-CS (*n* = 32) Mean (SD) | e-TAU (*n* = 30) Mean (SD) | *p* | Hedge's *g* (95% CI) |
|---|---|---|---|---|
| CAPS-5 total score | | | | |
| Pre-treatment | 37.56 (5.81) | 37.23 (5.91) | 0.89 | 0.06 |
| Post-treatment | 24.30 (9.55) | 27.59 (9.87) | 0.22 | 0.33 |
| 1-month f/up | 16.31 (10.59) | 20.44 (11.79) | 0.10 | 0.37 |
| 3-month f/up | 9.73 (9.61) | 19.00 (11.95) | <0.01 | 0.86 |
| PCL-5 total score | | | | |
| Pre-treatment | 52.63 (12.94) | 52.70 (12.32) | 0.95 | 0.01 |
| Post-treatment | 31.39 (17.03) | 36.18 (18.20) | 0.13 | 0.27 |
| 1-month f/up | 21.17 (17.61) | 27.63 (18.82) | 0.67 | 0.35 |
| 3-month f/up | 13.84 (15.02) | 25.20 (17.49) | 0.07 | 0.70 |
| DASS-Depression | | | | |
| Pre-treatment | 25.38 (10.56) | 25.53 (8.61) | 0.95 | 0.02 |
| Post-treatment | 11.29 (10.24) | 15.09 (10.81) | 0.13 | 0.36 |
| 1-month f/up | 10.32 (11.69) | 10.53 (10.50) | 0.67 | 0.02 |
| 3-month f/up | 4.58 (8.58) | 11.07 (11.44) | 0.07 | 0. 63 |
| DASS-Anxiety | | | | |
| Pre-treatment | 22.13 (9.68) | 22.47 (8.13) | 0.89 | 0.04 |
| Post-treatment | 13.93 (9.08) | 17.91 (10.54) | 0.13 | 0.41 |
| 1-month f/up | 10.32 (10.14) | 12.00 (10.54) | 0.51 | 0.16 |
| 3-month f/up | 6.92 (8.38) | 12.39 (11.54) | 0.15 | 0.61 |
| DASS-Stress | | | | |
| Pre-treatment | 27.31 (9.65) | 26.07 (7.13) | 0.62 | 0.15 |
| Post-treatment | 14.21 (9.89) | 19.45 (11.27) | 0.03 | 0.50 |
| 1-month f/up | 12.32 (11.24) | 12.42 (9.81) | 0.08 | 0.01 |
| 3-month f/up | 7.00 (8.02) | 14.40 (13.40) | 0.03 | 0.71 |
| Proportion of participants meeting clinical cut-off | | | | |
| CAPS-5 (<23) | *n (%)* | *n (%)* | *p* | |
| Post-treatment | 12 (37%) | 6 (20%) | 0.24 | |
| 1-month f/up | 18 (56%) | 9 (30%) | 0.17 | |
| 3-month f/up | 22 (69%) | 8 (27%) | 0.05 | |
| PCL-5 (<33) | *n (%)* | *n (%)* | *p* | |
| Post-treatment | 16 (50%) | 11(37%) | 0.58 | |
| 1-month f/up | 20 (62%) | 11 (37%) | 0.18 | |
| 3-month f/up | 23(72%) | 10 (33%) | 0.13 | |
| DASS-Depression (≤13) | *n (%)* | *n (%)* | *p* | |
| Post-treatment | 17 (53%) | 11 (37%) | 0.45 | |
| 1-month f/up | 20 (62%) | 11 (37%) | 0.10 | |
| 3-month f/up | 22 (69%) | 11 (37%) | 0.45 | |
| DASS-Anxiety (≤9) | *n (%)* | *n (%)* | *p* | |
| Post-treatment | 10 (31%) | 5 (17%) | 0.25 | |
| 1-month f/up | 13 (41%) | 11 (37%) | 0.78 | |
| 3-month f/up | 19 (59%) | 8 (27%) | 0.45 | |

*(Continued)*

**Table 5.** (*Continued*)

| Variable | PTSD Coach-CS (*n* = 32) Mean (SD) | e-TAU (*n* = 30) Mean (SD) | *p* | Hedge's *g* (95% CI) |
|---|---|---|---|---|
| DASS-Stress (≤18) | *n* (%) | *n* (%) | *p* | |
| Post-treatment | 22 (69%) | 11 (37%) | 0.02 | |
| 1-month f/up | 19 (59%) | 13 (43%) | 0.47 | |
| 3-month f/up | 22 (69%) | 10 (33%) | 0.26 | |

were overall very satisfied with the app (Item 15: M = 3.62; SD = 0.62) (see Supplementary Table S1). The lowest rating endorsed was for the help to learn about treatments for PTSD section (Item 2: M = 2.96; SD = 0.92). Providing a way for them to talk about what they have been experiencing was rated the highest (Item 14: M = 3.67; SD = 0.48).

Self-efficacy managing PTSD SYMPTOMS. Overall participants reported that they felt moderately to highly certain about their abilities to manage PTSD symptoms (see Supplementary Table S2). The lowest rating was for the ability to make themselves feel less sad or hopeless (Item 5: M = 65.56; SD = 29.80). The highest rating was for participants feeling that they can use the skills from the PTSD Coach app to manage their PTSD symptoms (Item 9: M = 81.86; SD = 22.71).

## Discussion

We set out to evaluate the feasibility and effectiveness of a four-session PTSD Coach-CS intervention for adults with PTSD in a South African resource-constrained setting. To our knowledge, this is the first time the PTSD Coach app was evaluated in an LMIC setting and was novel in supplementing intervention delivery with less specialised mental health services (registered counsellors).

We found a significantly greater reduction in PTSD symptom severity over time for the intervention versus control group for the primary outcome, the clinician-monitored PTSD symptom severity (CAPS-5). However, possibly due to the study being underpowered, we were not able to establish the superiority of the PTSD Coach-CS intervention at post-treatment, as hypothesised. The main effect was predominantly driven by significant between-group differences at 3-month follow-up and may point towards a potential lag in intervention effect and/or the benefit of longitudinal engagement with the app on greater symptom reduction over time. Interestingly, recent findings of the largest PTSD Coach-focused RCT (*N* = 234) following a similar procedure (four-session clinician-supported PTSD Coach app intervention) also did not find significant between-group effect size differences based on clinician-monitored PTSD symptoms at post-treatment (Possemato et al., 2023). Other trauma-focused therapies such as cognitive behavioural therapy and prolonged exposure therapy are typically longer in duration and may explain the delayed effect if participants continued to use the app on their own (Rossouw et al., 2018; Kaminer et al., 2023).

It should be noted that participants in the control group also improved over time, possibly due to the potentially beneficial effects of the monitoring sessions (clinician and counsellor); particularly since participants in the control group had high levels of trauma exposure, and the majority did not receive treatment during the study treatment period (Appelbaum et al., 2004; Grant et al., 2021; Korhonen et al., 2022). Receipt of a PTSD diagnosis, accompanying

symptom identification, and consequent monitoring could have normalised experiences for the first time and provided therapeutic benefit (Appelbaum et al., 2004; Whitworth 2016). However, further investigation in an adequately powered RCT is necessary to confirm this.

Self-reported PTSD symptom improvement over time was not significantly different between the groups. Possemato et al. (2023) found the inverse in their sample with significant between-group effects observed in the intervention group for self-reported PTSD symptoms and not for clinician-monitored symptoms. While notable, the discrepancy between clinician-monitored and self-reported PTSD symptom improvement is well documented in PTSD-focused research, with higher self-reported symptoms compared to clinician-monitored symptoms generally documented (Kramer et al., 2023; Resick et al., 2023). Reasons for this discrepancy include: comprehension of symptoms, non-trauma-related symptom endorsement and incorrect time-frame reference (i.e., not past month) (Kramer et al., 2023). In the present study, clinician observations during monitoring sessions support the former as being likely in our sample. However, further objective investigation is needed to support this notion.

We did not find a significant effect on self-reported depression and anxiety symptoms over time; despite previous PTSD Coach app research indicating the benefit of the app on depression symptoms (Kuhn et al., 2014; Possemato et al., 2016; Tiet et al., 2019; Hensler et al., 2022). To the best of our knowledge, the direct effect of the PTSD Coach app on anxiety has not been evaluated. However, we found a significant effect on self-reported stress symptom reduction over time and at post-treatment and at 3-month follow-up in the intervention group. This is not surprising given the nature of the PTSD Coach app management tools. Respectively, the deep breathing, muscle relaxation and mindfulness strategies (grounding, body scan, observing thoughts) included in the content of the app have demonstrated significant intervention effects on stress reduction (Goldsmith et al., 2014; Hopper et al., 2019; Liu et al., 2022). These findings may be related to longer-term app engagement and the possibility of a delayed intervention effect on PTSD symptoms mediated through reduced general stress symptoms.

The uptake of the intervention was good with all but one participant attending all intervention sessions and with app use outside the sessions varying from daily to five times per week. Participants perceived the PTSD Coach app as moderately to very helpful and were overall very satisfied with the app, findings similar to prior research (Kuhn et al., 2017; Pacella-LaBarbara et al., 2020; Hensler et al., 2022). Additionally, the positive reported self-efficacy in managing PTSD symptoms at post-treatment in our sample was similar to previous findings (Pacella-LaBarbara et al., 2020). The results support the usefulness of the app in our setting.

Preliminary concerns about the implementation of the intervention in a resource-constrained setting, for example, smartphone ownership, was not a significant barrier to intervention delivery in

our setting while it has been for other smartphone-based intervention studies (Kuhn et al., 2018; Sinha Deb et al., 2018; Bommakanti et al., 2020; Potdar et al., 2020). However, initial technical difficulties with downloading the app posed a potential barrier. While we encountered this during pilot testing, other PTSD Coach app researchers did not note significant technical challenges hindering access to the app (Bröcker et al., 2022). Language proficiency (English) was also not a significant barrier likely due to the high rate of bilingualism in South Africa, as well as the counsellor being available to assist if needed (Statista Research Department, 2023). Relatedly, the registered counsellor was effective in intervention delivery and risk management, thus findings support upscaling and task-shifting efforts with less specialised mental health services to broaden access to care in resource-constrained settings (Spedding et al., 2015; Singla et al., 2017; De Kock and Pillay 2018; Rossouw et al., 2018).

Limitations of the study include, first, a failure to reach the target sample size was caused by initial methodological challenges (Bröcker et al., 2022) and restrictions during the acute phase of the COVID-19 pandemic in South Africa, resulting in delayed initiation of the RCT. Additionally, recruitment and enrolment of the RCT itself were negatively impacted by systemic challenges related to loadshedding (i.e., rolling electricity blackouts) and regular public transport protests prohibiting study visit attendance. As such, the study was underpowered. Our post hoc power analysis, using the number of completers in each treatment arm and the between-group effect size detected at post-treatment confirmed that the study was underpowered (Power (1-β err prob) = 0.72). Moreover, a sample size calculation based on the present study data indicated that ($N = 102$) would be required to detect a significant between-group effect size at post-treatment ($\alpha = 0.05$; power = 0.80) (calculations were done using GPower v 3.1.9.7) (Faul et al., 2007).

Second, our comparator intervention (e-TAU) arm largely constituted a waitlist group as the majority of these participants did not receive clinical care during the trial period, further demonstrating the unfortunate reality of usual care in the South African setting. Future RCT studies should consider active comparators (e.g., a counsellor-supported versus self-managed PTSD Coach app intervention).

Third, study procedures included the provision of transport costs (to attend sessions) and data costs (for app download) therefore limiting the extent to which we can comment on the financial feasibility of the PTSD Coach-CS intervention in the South African context where many may not be able to afford these costs. Future research should consider replication of the study intervention procedures, provided by registered counsellors trained in the intervention protocol, at community clinics where the majority of South Africans receive routine or usual care. This approach can address transport barriers and may further assist with the feasibility and acceptability of the original PTSD Coach app intervention in the South African setting, where counsellors can assist with app installation and with crossing cultural and language barriers when needed. Future studies should also consider flexibility in intervention delivery procedures by conducting procedures virtually when needed (e.g., during violent public transport protests when travel using these means is unsafe) to further optimise intervention delivery in a real South African setting. Fourth, although self-reported app usage was high, we did not use the research version of the app and, therefore, could not include objective app usage in the analysis. Monitoring app use in future research may provide valuable information and may explain longer-term findings if app use continues over a prolonged period. Relatedly, future research could consider contextual adaption of the app to the applicable setting. In the South African setting, with 12 official languages, the use of counsellors to contextualise experiences and assist with language barriers when needed may be a more feasible and better use of resources. The latter is supported by the overall positive reception and potential effect of the intervention observed in our more diverse sample despite using the original version developed for veterans in the United States. Finally, the predominantly female representation in our sample limits the generalisability of our findings to the male population in South Africa. While our sample aligns with the overall higher documented PTSD prevalence among females and accompanying treatment-seeking behaviour, future research could aim for a higher male representation (Olff, 2017). While the largely black and isiXhosa representation in our sample can be seen as a limitation further influencing generalisability, it is also a strength as it further supports the feasibility of the original version in our setting with counsellor support to bridge cultural and language barriers.

Despite the above limitations, our results are promising and support further research that should consider methods to further bridge treatment barriers in our setting (i.e., evaluating the intervention at primary health care clinics in the community to limit transport difficulties). Incorporating the PTSD Coach app into usual care could also be explored to offer support while patients are on waiting lists, or to be of support between sessions and after receipt of care (e.g., termination of therapy).

In summary, this is the first study to evaluate the PTSD Coach app intervention in a resource-constrained setting, supplemented with less-specialised mental health services, and our findings contribute to the growing research on internet-based interventions. Based on PTSD symptom improvement over time, our findings suggest that a low-cost counsellor-supported PTSD Coach app intervention is a feasible, suitable and potentially effective treatment alternative for adults with PTSD in a resource-constrained setting. Originally developed for veterans in the US, our findings indicate that use of the app in a culturally different setting is plausible, but that more research in larger samples is needed to fully establish the effectiveness of the PTSD Coach-CS intervention in reducing PTSD symptom severity.

**Open peer review.** To view the open peer review materials for this article, please visit http://doi.org/10.1017/gmh.2023.92.

**Supplementary material.** The supplementary material for this article can be found at https://doi.org/10.1017/gmh.2023.92.

**Data availability statement.** The data that support the findings reported in this study are not shared publicly for reasons of privacy; however, they can be made available from the corresponding author (E.B.) upon reasonable request.

**Acknowledgements.** This project and resultant findings would not have been possible without the participants who travelled to attend sessions, shared their experiences and opinions and ultimately contributed to knowledge to inform appropriate interventions in our setting.

**Author contribution.** E.B. was the doctoral student who led the study with respect to conceptualisation, design and conduct of the RCT, data analysis and interpretation and drafting and revision of the manuscript. L.G. was the counsellor involved and was instrumental in intervention delivery and data collection. M.K. was the statistical expert and consultant who assisted with data analysis and interpretation. M.O., S.Su and S.Se provided extensive input into the conception and design of this RCT and the manuscript. All authors approved this manuscript for publication.

**Financial support.** E.B. is supported by the National Research Foundation Thuthuka Funding Scheme, the South African Medical Research Council (SAMRC) through its Division of Research Capacity Development under the SAMRC Bongani Mayosi National Health Scholars Programme of the South African National Treasury and the SAMRC through the Extramural Genomics of Brain Disorders Unit. S.Su was supported with funding from the SAMRC through a Self- Initiated Research Grant. S.Se is supported by the South African Research Initiative in PTSD, funded by the Department of Science and the National Research Foundation and the SAMRC Extramural Unit on the Genomics of Brain Disorders. The content herein is the sole responsibility of the authors and does not necessarily represent the official views of the SAMRC and other funders.

**Competing interest.** The authors declare none.

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
