## [Reviewer Report]

Dear Editor

We wish to submit two manuscripts for consideration by Cambridge Prisms: Global Mental Health. The first is titled ‘A clinician-monitored ‘PTSD Coach’ intervention versus enhanced treatment as usual in a resource-constrained setting: A randomised controlled trial’ and the second ‘Participants’ experience of a PTSD Coach-CS intervention in a resource-constrained setting.” These manuscripts align with the scope of the journal as we aimed to address treatment gaps by identifying and evaluating an intervention that may widen access to psychiatric care in a South African resource-constrained setting. We confirm that these manuscripts are original, have not been published, nor are they submitted elsewhere. 

In the first manuscript, we report on a randomised controlled trial (RCT) that evaluated the effectiveness of a four-session counsellor-supported PTSD Coach (PTSD Coach-CS) mobile application intervention in reducing PTSD, depression, anxiety, and stress symptoms in a South African adult community sample. The PTSD Coach mobile app is freely available, and supplementing delivery with counsellor support aimed to both increase intervention uptake and address upscaling of human resources to address the needs of an overburdened public healthcare system. The findings indicated symptom improvement in clinician-monitored PTSD symptoms and self-reported stress over time as well as good intervention uptake and satisfaction.

The second manuscript reports on the findings of a qualitative sub-study that formed part of the RCT procedures. These findings augment the quantitative results and provided valuable information to inform PTSD Coach-CS intervention amendments to increase its usefulness in a South African resource-constrained setting. 

This was the first time the PTSD Coach app was evaluated in a resource-constrained setting such as South Africa and the first to supplement intervention delivery with less specialised mental health services. Our findings are encouraging and support that a low-cost more accessible PTSD Coach-CS intervention is a feasible and potentially effective treatment alternative for adults with PTSD. Identification of alternative or additional treatment strategies is of particular importance based on the envisioned transition to the National Health Insurance (NHI) in South Africa. The transition towards NHI will provide quality affordable health services to all South Africans, irrespective of their socioeconomic status and is an important step towards narrowing longstanding treatment gaps. However, this may place an extra burden on an already stretched public health care system and the availability of low-cost scalable interventions can mitigate this burden. 

We hope that our submission will be favourably considered.

---

## [Reviewer Report]

The study makes a potentially valuable contribution to the literature on treating PTSD in lower-resources settings. A novel aspect is the use of a non-specialist provider to support participants using the PTSD Coach digital app. On the whole the manuscript is clearly written, however the rationale for the study needs further elaboration and more information is needed on the study methods. My recommended revisions are listed below.

Abstracts:

I’m not sure if the brief abstract will be used for publication but if so it needs a few minor adjustments:

1) indicate which were primary vs secondary outcome measures

2) It should be PTSD Checklist rather than PTSD checklist

3) Give the DASS-21 acronym after the full-scale name

4) Note that a survey of user experience was also conducted

For the extended abstract, the survey of user experience should again be noted under Methods.

The Introduction needs some re-working to provide a stronger and clearer rationale for the study:

1) The Introduction begins with noting the structural and attitudinal barriers to treatment access in South Africa and other LMICs. Yet the intervention being tested requires in-person session attendance, so it’s not clear how this actually addresses the structural barriers. If the claim is that the intervention will increase treatment access, the authors need to explain how it will do so (e.g. needing to attend fewer sessions to obtain an effective ‘dose’, because of the additional between-session use of the app?). It’s also not clear how the intervention can reduce stigma and other attitudinal barriers to help-seeking, since seeing a counsellor face-to-face is required. Relatedly, it’s not really clear why an in-person, counsellor-supported version of PTSD Coach was selected, when a self-managed version, or one that uses telephonic contact with a counsellor, may have circumvented many of the treatment barriers mentioned. There is only one sentence justifying this choice, and this is based on research in better resourced settings. Elaboration of how the advantages of in-person counsellor support can potentially balance the burden and stigma of attending in-person sessions is needed.

2) It’s not clear in the Introduction whether there are other mobile apps available to support people with PTSD and whether the benefits of PTSD Coach exceed those of the other apps (e.g. are other apps more costly? Do they have less evidence to support their effectiveness?). It would also be useful to have some information the background of PTSD Coach: where was it developed, when was it launched, who is aimed at (adults with PTSD, posttraumatic stress symptoms, comorbid depression/anxiety? People with PTSD who are in treatment or not in treatment?).

3) The sentence “Additional benefits include enhancements for visual and hearing impairments and existing evidence-based research that continues to expand” needs references for both these points.

4) The Manage components of the app need more description – what do these focus on? This is briefly noted in Table 1 but the reader needs some orientation earlier. Are there any trauma-focused elements, eg any components that address intrusive symptoms, negative trauma-related beliefs or behavioural/cognitive avoidance? In the Intro it is not really clear how the app addresses PTSD symptoms, as opposed to just being a general mental health/stress support app.

Some more detail should be added to the Method section:

1) Was an existing Coach PTSD – CS treatment protocol used (eg the one used by Possemato et al.?).

2) For participant recruitment, it states that flyers distributed where in the community. It would be useful to have some more information about what sorts of community sites the flyers were distributed to, as it is not really clear where participants were recruited from.

3) Under Measures, add info on whether the measures have been used and/or validated in SA previously.

4) Under Interventions, make it clear that participants attended in-person sessions and give information about the study site/session (a clinic? hospital? tertiary institution?).

5) Explain what a registered counsellor means, for audiences outside South Africa.

6) What does it mean that participants “worked through selected symptoms and tools with the counsellor to assist with technical difficulties”? It’s not really clear if the counsellor offered any psychological support or coaching on practicing the tools, or just gave instructions about the technical aspects of accessing and implementing the tools. It would be helpful to understand more about what the counselling relationship entailed and what the role of the therapeutic relationship might have been in producing a therapeutic effect.Measures

7) At the follow-ups, did you check if participants had been using the app post-treatment?

Results

The results are clearly presented. In Table 4, there seems to be an error in the row for app use over past week. For the last session, 24 of the 28 who attended sessions used the app – this would be 86%, not 36%?

Discussion

1) It should be noted that the provision of transport costs and data costs for app download limits the degree to which you can comment on the feasibility of PTSD Coach–CS in the SA context, where many people may not be able to afford these. It would be interesting to test how PTSD Coach-CS compares with self-managed PTSD Coach or PTSD Coach with telephonic/whattsapp counsellor support. If these modalities are also effective (the Possemato et al. study found that self-managed PTSD Coach was still effective, although less so than the CS version), this could address the contextual barriers to transport and treatment attendance that contributed to this study being underpowered, hence balancing feasibility with effectiveness.

2) Under limitations, note that the TAU group was largely a waiting list or inactive control group (which is, unfortunately, the nature of usual treatment in South Africa), as two thirds received no active support, and that a larger RCT would need to consider more active control conditions (eg self-managed PTSD Coach).

3) The authors state “Due to the study being underpowered, we were not able to establish superiority of the PTSD Coach-CS intervention at post-treatment.” Failing to find a significant treatment effect at post-treatment MAY have been due to the study being under-powered, but we cannot know for certain that this was the case – there may have been no significant effect even with a larger sample. This needs to be re-phrased.

---

## [Reviewer Report]

Thank you for the opportunity to review an well-written manuscript. The reviewer has appreciation for the many challenges, including COVID in completing this research.

I have no requirement for any changes to any specific aspect of the research design or the reporting of the results.

Whilst I do not see the need for any changes prior to publication, possible thoughts I had, include:

Report on the impact of a greater loss in the control group on statistical analyses of the between group data.

In the introduction and discussion perhaps discuss the impact of the type of intervention. It is the reviewers view that an intervention that included an trauma-focused intervention component that focused on identifying triggers and allowed some type of practice of remembering and retelling the trauma would have improved the outcome. I am wondering to what degree this was covered in the manage symptoms, coping with triggers, observe thoughts tools and how much time was spent on these components, compared to a longer intervention. Also given the role of avoidance in maintaining PTSD, to what degree would the intervention encourage participants to allow themselves to think of (trauma memory) and face triggers as a result of the intervention.

It would also be interesting to understand the extent of the module on sleep and to what degree any change in sleep routine might have also contributed to an improvement in PTSD.

Congratulations on well-executed research and excellent reporting of the results.

---

## [Reviewer Report]

Please view the letter titled ‘Response to reviewers 1 and 2_19.11.2023’ submitted with the revised manuscript.

---

## [Reviewer Report]

The revised manuscript has addressed my recommendations and the manuscript is suitable for publication.